# The Effect of Mediterranean Diet on Cognitive Functions in the Elderly Population

**DOI:** 10.3390/nu13062067

**Published:** 2021-06-16

**Authors:** Blanka Klimova, Michal Novotny, Petr Schlegel, Martin Valis

**Affiliations:** 1Department of Applied Linguistics, Faculty of Informatics and Management, University of Hradec Kralove, 50003 Hradec Kralove, Czech Republic; 2Department of Neurology, University Hospital Hradec Kralove, 50005 Hradec Kralove, Czech Republic; novotmi1@fnhk.cz (M.N.); martin.valis@fnhk.cz (M.V.); 3Department of Physical Education and Sports, Faculty of Education, University of Hradec Kralove, Rokitanskeho 62, 50003 Hradec Kralove, Czech Republic; petr.schlegel@uhk.cz

**Keywords:** Mediterranean diet, older people, cognitive impairment, impact

## Abstract

At present, due to the demographic changes and the rise of senior population worldwide, there is effort to prolong an active life of these people by both pharmacological and non-pharmacological strategies. The purpose of this article is, on the basis of the literature review of recent clinical studies, to discuss one of such strategy, i.e., the effect of Mediterranean Diet (MedDiet) on the cognitive functions among both the cognitively unimpaired and impaired elderly people. The methodology includes a literature review of full-text, peer-reviewed journal studies written in English and published in Web of Science and PubMed between 1 January 2016 and 28 February 2021. The findings indicate that the adherence to MedDiet has a positive effect on both cognitively impaired and unimpaired older population, especially on their memory, both in the short and long run. The results show that the higher adherence to MedDiet proves to have a better effect on global cognitive performance of older people. In addition, the adherence to MedDiet offers other benefits to older people, such as reduction of depressive symptoms, lowered frailty, as well as reduced length of hospital stays.

## 1. Introduction

Globally, there is a sharp rise in the elderly population. In fact, in 2019 there were 703 million of the elderly at the age of 65+ years. By 2050, this number should be more than doubled and reach 1.5 billion [1]. Due to this demographic aging of the population, there is a higher incidence of worsen health conditions, such as impaired vision, hearing, but also a progressive decline of cognitive functions [2].

Cognitive functions are all mental processes that allow us to recognize, remember, learn and adapt to ever-changing environmental conditions. This includes, for example, learning, memory and thinking, receptive functions, such as perception of stimuli, their maintenance and sorting, as well as expressive functions, such as speech, writing, and drawing [3]. These are all symptoms of a neurodegenerative disorder known as dementia, associated with a loss of cognitive abilities [4]. Dementia is a progressive disorder, which gradually limits a persons’ autonomy and self-sufficiency, and in later stages it might result in their disability [5]. Unfortunately, at the moment, it cannot be cured, but it can be only delayed by both pharmacological and non-pharmacological therapies [6]. The pharmacological therapies include four main drugs, such as acetylcholinesterase inhibitors (donepezil, galantamine, rivastigmine) or memantine (also an N-methyl-D-aspartate (NMDA) antagonist) and they can only improve the mental state of the affected person temporarily and only slow down the pathological process [7].

Therefore, it is necessary to try to prevent the development of brain degeneration. At present, there exist several non-pharmacological therapies, which have been proved to prevent the development of cognitive impairment in old age [8]. These involve physical activities, which should be done regularly at least three times a week, cognitive training, for example, solving crosswords, reading, or learning a foreign language, and a healthy diet [9], out of which Mediterranean diet seems to be a solution [10] since evidence suggests that greater adherence to Mediterranean diet (MedDiet) is associated with slower cognitive decline and lower risk of developing Alzheimer disease [11,12]. Although the exact mechanism by which MedDiet positively affects cognitive function remains unknown, the findings suggest that this might be a multifactorial process [13]. The most important active molecules with a protective effect on nervous tissue or metabolism involve the influence of saturated fatty acids and the anti-inflammatory and antioxidant action of the whole group of active biomolecules present in this type of diet. The fact that the MedDiet is associated with caloric restriction and balanced nutrient intake, which in turn positively influences the overall physiological metabolic processes in the body (e.g., insulin resistance, blood glucose levels and lipid profile), also has a significant impact [14,15]. MedDiet consists of a large number of vegetables, fruits, beans, legumes, whole grains, olive oil, seeds, herbs and spices, as well as fish, seafood, eggs, cheese and poultry and disrecommends unusual red meats and sweets. In particular, a high intake of fish and low intake of alcohol contributes to the delay of cognitive decline [11].

Research indicates that adherence to MedDiet during midlife is associated with 36%–46% greater likelihood of healthy ageing [16,17,18]. For instance, Critselis and Panagiotakos [18] state that adherence to MedDiet among the elderly people is highly associated with healthy ageing since diets similar to that of MedDiet are connected with 269% greater likelihood of successful ageing and 33% reduction in mortality risk. Furthermore, Buglio et al. [19] maintains that MedDiet has a positive impact on hospitalized patients, i.e., it slows down their rate of stay length and in-hospital mortality.

The purpose of this article is, on the basis of the literature review of recent clinical studies, to discuss the effect of MedDiet on the cognitive functions among both the cognitively unimpaired and impaired elderly people.

## 2. Materials and Methods

The authors of this review followed the Preferred Reporting Items for Systematic Reviews and Meta-Analyses (PRISMA) guidelines. They systematically reviewed English written research studies from peer-reviewed journals and published in Web of Science and PubMed between 1 January 2016 and 28 February 2021 in order to discuss the most recent findings regarding this research topic. Only randomized clinical trials and cohort studies were considered. The main focus was the impact of MedDiet on cognitive functions among elderly people. The identified studies included groups where the population had to be 55+ years old. The searched collocation keywords were as follows: Mediterranean diet and cognition, Mediterranean diet and dementia, Mediterranean diet and Alzheimer’s disease, Mediterranean diet and cognitive decline, Mediterranean diet and the elderly.

All the authors conducted the quality evaluation of these articles based on Health Evidence Quality Assessment Tool for review articles.

The primary results of this review aim to:Assess and discuss the recent randomized clinical trials (RCT) and cohort studies dealing with the impact of healthy diet on cognitive performance in both cognitively unimpaired and impaired seniors;Discuss the most recent findings regarding this research topic.

Altogether, 475 studies were detected in the Web of Science and PubMed. A total of 241 studies were found in PubMed and 234 articles were identified in the Web of Science. Another 3 studies were generated from other sources; generally, from the references of the detected articles. After excluding duplicates and titles/abstracts not connected with the research issue, 109 articles remained. Of these, only 43 articles were suitable for the research issue because the rest of the detected articles (76), after screening their content, were found irrelevant. Therefore, only 43 studies were examined in full, and they were considered against the following inclusion and exclusion criteria. The inclusion criteria were as follows:The review period was limited to 1 January 2016 and 28 February 2021.The articles had to be published in peer-review English written journals.Only randomized controlled trials and cohort studies were included into the review.The primary results had to focus on the impact of MedDiet on cognitive functions among elderly people.The subjects had to be at the age 55+ years old.The exclusion criteria were as follows:The review studies [10,16,17,18], slightly different research focus [19], multi-domain intervention studies [9,20,21], study protocols [22,23], the studies outside the determined period [11,12] or the studies with different age of participants [24].

Considering the above-described criteria, eight studies were eventually included into the final analysis. Figure 1 below describes the selection procedure.

## 3. Results

Altogether eight original studies [25,26,27,28,29,30,31,32] were detected. Five studies were randomized controlled trials [26,29,30,31,32] and three [25,27,28] were cohort studies. Three of the studies originated outside the Mediterranean area [27,31,32] and one partly [29]. The results confirm that at least one group adhered to MedDiet. The research samples consisted between 35 and 2092 subjects. In four studies [25,26,28,31], the subjects were cognitively impaired and suffered from Alzheimer’s or Parkinson disease. In the remaining studies, the participants were healthy older individuals. The intervention period ranged between three weeks and one year. The main outcome measures included a battery of standardized cognitive tests, but also different kinds of other tests focused on nutrition/diet assessments or biochemical evaluation aimed at body mass index.

Overall, the strength of the findings of the detected studies is that the adherence to MedDiet improves memory of both cognitively unimpaired and impaired older people.

The main weakness of this study is the use of different outcome measures, as well as varied research samples and intervention periods, as well as measured target domains.

Table 1 below provides an overview of the mains findings from the detected studies. The findings are discussed in alphabetical order by their first author.

## 4. Discussion

The findings described above indicate that the adherence to MedDiet has a positive effect on both cognitively impaired and unimpaired older population, especially on their memory, both in the short and long run. In addition, the higher adherence to MedDiet proves to have a better impact on global cognitive performance of older people [25,27,28,29,30], which has been also evidenced in other research studies on the same issue [33,34]. 

Furthermore, Mantzorou et al. [28] expands that higher adherence to MedDiet was significantly associated with healthy younger age, female gender, higher educational level, and better anthropometric parameters. These findings are in line with other studies, e.g., Okubo et al. [35], who report that females are more concerned about their selection of individual nutrients since the dietary approach. In their study, they also discovered that especially females with a higher educational degree, going outdoors frequently and avoiding smoking and drinking alcohol had a better cognitive performance than the others. Overall, females of any age do care about their diet more than males of the corresponding age [27]. Moreover, de la Rubia et al. [26] confirm that this is also partly true for cognitively impaired female patients with AD, who recover more easily than male patients when exposed to MedDiet.

The results from the studies performed among Alzheimer’s disease (AD) and Parkinson’s disease (PD) patients and focused on the association of MedDiet on their cognitive functions reveal that this dietary pattern has a positive impact on the following cognitive domains: episodic, temporal orientation, semantic memory, language, attention, or concentration [26,32].

In addition, the findings of this review indicate that enrichment of MedDiet with a higher dosage of some food, such as coconut oil [26], extra-virgin olive oil (EVOO) [30], or fresh, lean pork [32], might have a more significant impact on the improvement of cognitive performance among seniors than just MedDiet alone.

Research shows that particularly polyunsaturated fatty acids and flavonoids play an important role in the enhancement of cognitive performance among healthy older people [36,37]. In fact, fatty acids form the main components of the neurobiomembrane and thus interfere significantly in processes, such as nerve signal transduction and neurotransmission at synapses [38]. For example, Gu et al. [39] found that adequate intake of PUFA from fish was positively associated with gray matter volume in patients with Alzheimer’s dementia. Strike at al. [40] reported that PUFAs (omega-3) improved white matter integrity and processing speed. Most prospective cohort studies on this topic [41,42] are relatively uniformly positive about the effect of fish consumption on cognitive decline. A neglected but nevertheless interesting molecule appears to be the antioxidant carotenoid astaxanthin. This biomolecule is produced by algae and is responsible for the dark red-orange color of salmon, shrimp and lobster meat. Astaxanthin crosses the blood-brain barrier and has effects on the central nervous system, including antioxidant, anti-inflammatory and antiapoptotic effects [43].

In addition, extra-virgin oil, particularly its component, secoiridoid oleuropein, might decrease the risk of cognitive decline [44]. There are not many studies dealing directly with the effect of EVOO on cognition [30]. The effect of low doses over short periods of time has not been studied at all, however, there are studies that have studied the protective effect of EVOO on brain structures over long periods of administration (e.g., 6.5 years) [45]. In the literature, EVOO supplementation has been reported to affect tests of cognitive function, even at low doses [45,46,47], while MedDiet containing EVOO may prevent cognitive decline over long periods of time and have a beneficial effect during the long prodromal phase of dementia [48].

Generally, a wide variety of anti-inflammatory and antioxidant substances form a very important group of MedDiet nutrition in terms of preserving human cognitive abilities. These include vitamins (A to E), folic acid, phenolic flavonoids (especially oleuropein) [44] and biomolecules of a lipophilic nature (especially alpha-tocopherol, beta-carotene) [29,30]. MedDiet also contributes to reduce symptoms of other chronic diseases, such as autoimmune diseases or cardiovascular diseases [49].

In addition to the classic MedDiet, which does not have a strictly defined calorie intake, calorie-deficient variants have also been tested. For example, Gepner et al. [50] combined the MedDiet and a low-carb diet, which reduced carbohydrate intake (initially < 40 g/day, then <70 g/day) and increased protein and fat intake. Another characteristic of this Med/LC diet was the low red meat content, with poultry and fish replacing beef and lamb.

Present research indicates that overall, a higher varied diet consisting of relevant dietary nutrients, such as MedDiet, and no single foods, has a more far reaching impact on cognitive performance of older inhabitants [27,29,51]. There are also other healthy diets, similar to MedDiet, which in fact reflect regional dietary habits. These include, for example, Nordic diet (NPDP) [52] or MedÉire diet [53], both emphasizing traditional, sustainable, and locally sourced foods in order to better adopt and adhere to MedDiet dietary patterns among these people. Shakersain et al. [52] in their study showed that moderate-to-high adherence to the NPDP may predict a better-preserved cognitive function among older adults in Nordic countries in comparison with neurodegenerative delay, Mediterranean diet, dietary approaches to stop hypertension, and the Baltic Sea diet.

However, there are also studies, which find no association between the adherence to MedDiet and better cognitive performance among elderly [54,55]. This is, for example, true for the study by Hill [56], whose study shows that there is no connection between the adherence to MedDiet and beta-amyloid deposition in a cohort of healthy Australian women.

In conclusion, the adherence to MedDiet has a positive impact on both cognitively impaired and unimpaired older population. Furthermore, it brings other benefits, such as reduction of depressive symptoms, lowered frailty, as well as reduced length of hospital stays.

Future research should focus on more clinical trials, which would confirm the potential role of MedDiet MD in reducing the risk of cognitive impairment.

## Figures and Tables

**Figure 1 nutrients-13-02067-f001:**
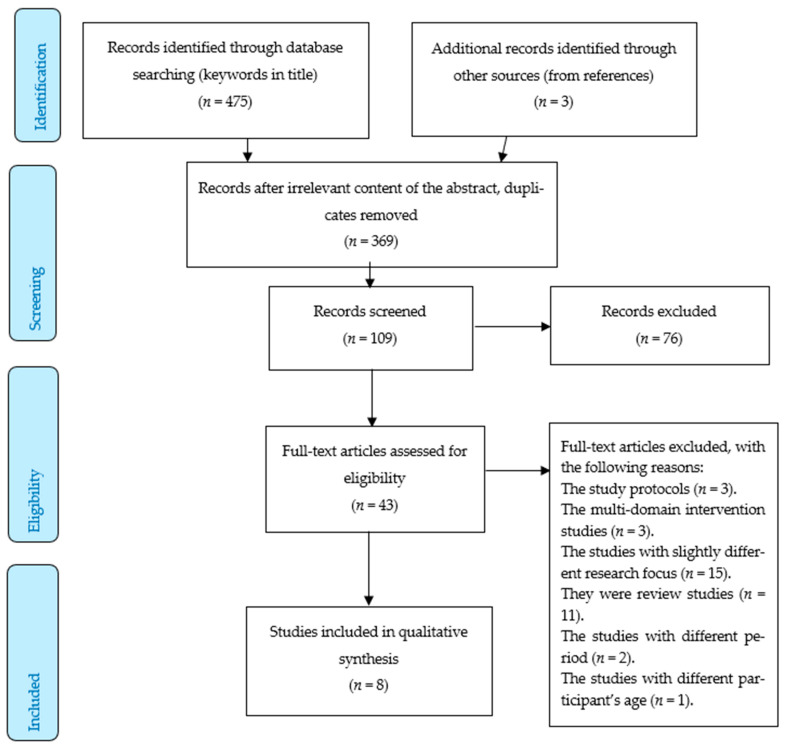
An overview of the selection procedure.

**Table 1 nutrients-13-02067-t001:** An overview of the results.

Author and Type of the Study	Aim of the Study	Characteristics of the Research Sample and MedDiet	Outcome Measures	Findings
de Amicis et al. [25] cohort study-cross-sectional Italy	To examine the impact of MedDiet on cognitive performance of older Italian people.	279 subjects, aged 65+ years (80 men, 199 women), conducted between June 2015 and December 2016.	14-item questionnaire, Mini-Mental State Examination (MMSE).	The MedDiet is associated with a lower risk of cognitive impairment (odds ratio (OR) = 0.39; 95% confidence interval (CI), 0.15–0.99; *p* = 0.045).
de la Rubia et al. [26] RCT Spain	To identify modifications in the main cognitive functions of patients with AD after following a coconut oil enriched Mediterranean diet.	44 patients with AD, aged 65 to 85 years old, an experimental group (22 patients) followed a coconut oil enriched Mediterranean diet, and a control group (22 patients) followed a Mediterranean-style diet. It lasted 21 days.	7 min screen, which analyses temporal orientation, visuospatial and visuoconstructive abilities, and semantic and episodic memory.	The experimental group improved in episodic, temporal orientation, and semantic memory and it seems that the positive effect is more evident in women with mild-moderate state, although other improvements in males and severe state were also shown.
Karstens et al. [27] cohort study-cross-sectional USA	To explore cross-sectional associations between the MedDiet and cognitive and neuroimaging phenotypes associated with AD and VaD (separately) among cognitively unimpaired seniors.	82 healthy seniors at the age of 68.8 years, 50% males and 50% females; participants were divided into high and low (median split) adherence groups.	Block Food Frequency Questionnaire 2005, standardized cognitive assessment battery of tests (i.e., the California Verbal Learning Tests, trail making tests—Part A, Part B, and the Wechsler Test of Adult Reading), MRI, T1-weighted images, FreeSurfer 6.0 segmentation pipeline.	The high MedDiet group was better at learning and memory performance (β = 0.52, SE = 0.21, t (74) = 2.53, *p* = 0.01, d = 1.23).
Mantzorou et al. [28] Cohort study Greece	To assess MedDiet adherence of older Greeks on their cognitive functions and mental state.	2092 males and females, both cognitively healthy and unhealthy, mean age 74.97 ± 8.41 years, from seven different Greek cities.	Mini-Mental State Examination (MMSE), Geriatric Depression Scale (GDS), and Mediterranean Diet Score (MedDietScore) questionnaires.	Higher MD adherence is strongly associated with better cognitive status and less depressive symptomatology.
Marseglia et al. [29] RCT China, France, Italy, Netherlands, Poland, Sweden, UK	To explore the impact of NU-AGE’s dietary intervention on age-related cognitive decline.	1279 healthy seniors, age range: 65–79 years, from five European centers, a control group 638, was adhered to MedDiet and an intervention group 641, was adhered to habitual diet.	CERAD—neuropsychological battery, MMSE, Babcock Story Recall Test, pattern comparisons, digit cancellation, trail making tests, word list memory, 15-items Boston Naming Test, Constructional Praxis Test, category fluency. Assessed cognitive domains: global cognition, perceptual speed, executive function, episodic memory, verbal abilities, and constructional praxis.	Subjects with higher adherence to the NU-AGE diet experienced considerable improvements in global cognition (β 0.20 (95% CI 0.004, 0.39), *p*-value = 0.046) and episodic memory (β 0.15 (95% CI 0.02, 0.28), *p*-value = 0.025) after 1 year, compared to those adults with lower adherence to NU-AGE diet. Both groups of subjects improved in global cognition and in all cognitive domains after 1 year.
Mazza et al. [30] RCT Italy	To examine whether the replacement of all vegetable oils with a lower amount of extra-virgin olive oil, in the contest of a Mediterranean Diet, would improve cognitive performances, among elderly Italian individuals.	110 participants, mean age was 70 ± 4 years; an experimental group had MedDiet in which all vegetable oils (including olive oil, high-oleic sunflower oil, high-oleic sunflower oil, canola oil and hydrogenated vegetable oils) were substituted by extra-virgin OO at dose of 20–30 g per day, and a control MedDiet alone. It lasted 1 year.	Neuropsychological tests (MMSE and ADAS-cog), anthropometric measurements and cardiovascular risk factors assessment, dietary intake data were assessed by a 24-h recall and a 7-day food record and calculated using nutritional software MetaDieta, biochemical evaluation.	A higher reduction of ADAS-cog scores (improved test) after 1 year in the seniors of the MedDiet plus low dose of extra-virgin OO group than that observed with a MedDiet alone (−3.0 ± 0.4 Vs. −1.6 ± 0.4 respectively).
Paknahad et al. [31] RCT Iran	To examine the impact of the Mediterranean diet on cognitive functions in patients with PD.	80 patients, mean age-60 years, the experimental group followed MedDiet (*n* = 40) or control (*n* = 40) group. It lasted 10 weeks.	Montreal Cognitive Assessment (MoCA) test, BMI test, Nutritionist IV software.	The mean score of the dimensions of executive function, language, attention, concentration, and active memory and the total score of cognitive assessment significantly increased in the intervention compared with the control group (*p* < 0.05, for all).
Wade et al. [32] RCT Australia	To explore the cognitive effects of MedDiet with additional red meat.	A 24-week parallel crossover design compared MedDiet with 2–3 weekly servings of fresh, lean pork (MedPork) and a low-fat (LF) control diet. 35 participants aged between 45 and 80 years and at risk of cardiovascular disease followed each intervention for 8 weeks, separated by an 8-week washout period.	Cambridge Neuropsychological Test Automated Battery, SF-36 Health Survey, Profile of Mood States.	Compared to LF, the MedPork intervention led to higher processing speed performance (*p* = 0.01) and emotional role functioning (*p* = 0.03).

Explanation: AD: Alzheimer’s disease, BMI: body mass index, MedDiet: Mediterranean diet, NU-AGE: new dietary strategies addressing the specific needs of elderly population for an healthy ageing in Europe, PD: Parkinson’s disease, RCT: randomized controlled trial, VaD: vascular disease.

## Data Availability

Not applicable.

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
