# Peer review of "The Effect of Mediterranean Diet on Cognitive Functions in the Elderly Population"

_nutrients, 2021, doi:10.3390/nu13062067_

Round 1

Reviewer 1 Report

This good & succinct paper shows the beneficial health effects of a meditteranean diet on cognitive function, an often-overlooked feature, as both prevalence and incidence of Alzheimer and vascular dementia increases. I do have the following remarks:

  • The MedDiet also exists in a low-carb form; has any difference with the regular MedDiet ever been demonstrated in this form?
  • Previously, a cognitive function study in Nutrients had compared the MedDiet to the Nordic Diet (Shakersain B, Nutrients, 2018), and the authors should dicuss this.
  • A meta-analysis was not performed, and the authors should discuss why.
  • Line 62: disrecommends.

Author Response

Dear Reviewer,

Regards,

Authors

Reviewer 2 Report

In this review article, the authors summarized the findings of eigth original articles showing a positive impact of the mediterranean diet on cognitive decline in the eldelrly.

In general, the article re-states the findings already published and does not add to the existing literature. The authors need to rethink the focus of their article as to elaborate upon what is known.

Author Response

Dear Reviewer,

Regards,

Authors

Reviewer 3 Report

This is a well-written review on the value of Med diet.

There is a typo in fig1 (eligibilty, xx = ?).

The authors have provided 45 references and I would suggest them to include more material in their discussion.

The discussion should be improved and specific reference to ingredients of Med Diet should be mentioned here, like EVOO, marine and dairy.

Currently, there is lot or research linking inflammation and Med Diet and the authors should include this in their discussion.

Suggested references:

https://pubmed.ncbi.nlm.nih.gov/30319088/

https://www.ncbi.nlm.nih.gov/pmc/articles/PMC7400632/

Happy to review a revised version of the MS.

Author Response

Dear Reviewer,

Regards,

Authors

Round 2

Reviewer 3 Report

The authors have improved the MS.